# Global and Northern-High-Latitude Net Ecosystem Production in the 21st century from CMIP6 experiments

Han Qiu[1*†], Dalei Hao[2], Yelu Zeng[1], Xuesong Zhang[3], Min Chen[1*]

[1]Department of Forest and Wildlife Ecology, University of Wisconsin-Madison, Madison, WI, USA
5 [2]Atmospheric Sciences and Global Change Division, Pacific Northwest National Laboratory, Richland, WA, USA
[3]USDA-ARS Hydrology and Remote Sensing Laboratory, Beltsville, MD 20705-2350, USA
[†]Now at Atmospheric Sciences and Global Change Division, Pacific Northwest National Laboratory, Richland, WA, USA

*Correspondence to*: Han Qiu (han.qiu@pnnl.gov) and Min Chen (min.chen@wisc.edu)

10 **Abstract.** Climate warming is accelerating the changes in the global terrestrial ecosystems and particularly those in the northern high latitudes (NHL, poleward of 50 °N), and rendering the land-atmosphere carbon exchange highly uncertain. The Coupled Model Intercomparison Project Phase 6 (CMIP6) employs the most updated climate models to estimate terrestrial ecosystem carbon dynamics driven by a new set of socioeconomic and climate change pathways. By analyzing the future (2015-2100) carbon fluxes estimated by ten CMIP6 models, we quantitatively evaluated the projected magnitudes, 15 trends and uncertainties of the global and NHL carbon fluxes under four scenarios plus the role of NHL in the global terrestrial ecosystem carbon dynamics. Overall, the models suggest that the global and NHL terrestrial ecosystems will be consistent carbon sinks in the future, and the magnitude of the carbon sinks is projected to be larger under scenarios with higher radiative forcing. By the end of this century, the models by average estimate the NHL net ecosystem productivity (NEP) as 0.54±0.77, 1.01±0 .98, 0.97±1.62, and 1.05±1.83 PgC/yr under SSP126, SSP245, SSP370 and SSP585, 20 respectively. The uncertainties are not substantially reduced compared with earlier results, e.g., the Coupled Climate-Carbon Cycle Model Intercomparison Project (C4MIP). Although NHL contribute a small fraction of the global carbon sink (~13%), the relative uncertainties of NHL NEP are much larger than the global level. Our results provide insights into future carbon flux evolutions under future scenarios and highlight the urgent need to constrain the large uncertainties associated with model projections for making better climate mitigation strategies.

## 1 Introduction

The global terrestrial biosphere is considered as a major carbon pool and a key player in the global carbon cycle. In the last decade (2011-2020), the terrestrial biosphere absorbs $CO_2$ from atmosphere at a rate of about 120 Pg C/year by vegetation photosynthesis and releases a similar amount of carbon back to the atmosphere through respirations from plant metabolism and microbial activities (i.e., autotrophic and heterotrophic respirations) in response to climate oscillations and disturbances-induced emissions, resulting in a land carbon sink of about 3.4 Pg C/year with an additional 1.6 Pg C/year loss due to land use change (Friedlingstein et al., 2020). However, these numbers of land-atmosphere carbon fluxes, especially the photosynthesis and respiration components, change over time in response to climate change and are associated with large uncertainties. For example, using trace gas measurements, Campbell et al., (2017) estimated a large increase in global terrestrial biosphere photosynthetic carbon uptake of 31% over the 20th century accompanied with rapidly rising $CO_2$ concentration and warming climate. This estimate however did not agree with many carbon/climate models. The global soil respiration carbon flux has also been found increasing in the past several decades, according to the analysis of a global soil respiration database, but the degree to which climate change affects the changes of heterotrophic respiration is highly uncertain (Bond-Lamberty et al., 2018). Besides the scientific importance of understanding the long-term feedbacks between the terrestrial biosphere and the climate system, it is also critical to track the changes of the global land carbon budget for making manageable climate mitigation policies as it is a key component of the global carbon budget and has been considered as an important approach to achieve carbon neutrality.

Particularly, as the host of the most Earth's permafrost soils, arctic ecosystems store twice the amount of carbon as in the atmosphere and play an important role in the global carbon budget (Schuur et al., 2015; Tarnocai et al., 2009; Zimov et al., 2006). During the last few decades, the temperature in northern high latitudes (NHL, poleward of 50 °N) regions has been rising particularly fast. The Arctic Circle (66.5-90 °N) has warmed more than 0.7 °C per decade since 1979, almost four times faster than the globe (Rantanen et al., 2022). Previously stored soil carbon is potentially labialized by permafrost thawing and enhanced decomposition of soil organic carbon due to a warmer climate (Belshe et al., 2012; Koven et al., 2011; Natali et al., 2014; Schaefer et al., 2011; Schuur et al., 2015; Schuur and Abbott, 2011). This shapes a positive climate feedback since the excessive carbon release would in turn stimulate climate warming (Koven et al., 2011; Schuur & Abbott, 2011; Zimov et al.,

2006). On the other hand, $CO_2$ fertilization combined with other favorable conditions could enrich plant growth and drive the expansion of vegetation, e.g., arctic tundra and boreal forest, in the Arctic region, which may enhance plant carbon uptake and photosynthesis productivity (Berner et al., 2020; Liang et al., 2018; Mekonnen et al., 2019; Myers-Smith et al., 2020; Sistla et al., 2013). Despite the prevailing greening signal observed in the NHL, regional browning or negative Normalized Difference

Vegetation Index (NDVI) trend was also observed (Lara et al., 2018; Phoenix and Bjerke, 2016). Disturbances such as fire are also increasing in frequency and duration in response to the warming climate change, and exerting impacts on vegetation dynamics including canopy structure and functioning which in turn affects photosynthesis and ecosystem respirations (Hu et al., 2015; Mekonnen et al., 2019; Whitman et al., 2018). These evolving and counteracting processes complicate the determinations whether the NHL ecosystem functions as a carbon source or sink and how this will be projected in the future.

Great uncertainties are revealed from evaluating results of multiple Earth system models (ESMs) in the NHL region, with some ESMs showing NHL ecosystems as a carbon sink while others indicating an opposite sign (Fisher et al., 2014; Friedlingstein et al., 2014; Qian et al., 2010). Moreover, inconsistent model structure, diversified process representations as well as uncertainties in data, external variables and parameterizations further compromise the confidence in predictions of ESMs (Bradford et al., 2016; Luo et al., 2016; Todd-Brown et al., 2013).

The Coupled model intercomparison project (CMIP) coordinated a series of comprehensive comparisons among a handful climate models and has become an essential element of international climate research (Eyring et al., 2016; Taylor et al., 2012). Building on the previous Atmospheric Model Intercomparison Project (AMIP), CMIPs have broadened its purposes and contributions to a wide range of disciplines to foster understanding of evolutions and changes of climate and its impacts on societal sectors from history, to present and future (Eyring et al., 2016). Yet, great uncertainties were revealed from previous

CMIPs' results and the spread of the model responses to climate sensitivity remains large (Collins et al., 2013). A primary scientific gap of previous CMIP experiments is how the radiative forcing pathways, resulting from anthropogenic activities or natural emissions, could be optimally estimated (Stouffer et al., 2017). More recently, the CMIP phase 6 (CMIP6) employed a number of the most updated global climate models and endorsed 21 individually designed MIPs to address various scientific questions (Eyring et al., 2016). Guided by the goals to facilitate integrated research on the impact of future scenarios over

natural and human systems, and to help quantify uncertainties in future projections based on multi-model simulations, the

Scenario Model Intercomparison Project (ScenarioMIP; (O'Neill et al., 2016) incorporate a broad range of future scenarios with various combinations of Representative Concentration Pathways (RCPs) which was initially adopted in CMIP5 and newer Shared Socioeconomic Pathways (SSPs). These integrations allow a comprehensive assessment of plausible future climate conditions covering a wide span of mitigation and adaptation options (Riahi et al., 2017; van Vuuren et al., 2014), and represent the most updated understanding of the climate change and carbon cycle in the next few decades (Eyring et al., 2016; O'Neill et al., 2016). The CMIP6 ScenarioMIP takes advantage of previous CMIP resources and makes advancements in two major updates: first, the climate models employed are more updated with better representations of underlying physical processes; and second, the models are driven by a new set of emission pathways and land use scenarios, i.e., SSPs generated by updated versions of Integrated Assessment Models (IAMs) with new conceptual designs of future societal development and evolution with different assumptions on the challenges to mitigation and adaptation to the climate change (O'Neill et al., 2016). The variety of SSPs and RCPs combinations also cover a broader range of air pollutant emissions which are supposed to bridge the gap of relatively narrow aerosol scenarios adopted in CMIP5 (Stouffer et al., 2017).

The goal of this study is thus to answer the following questions based on the CMIP6 ScenarioMIP results: a) What is the future trajectory (spatial and temporal patterns) of global and NHL terrestrial carbon fluxes, in particular the net flux between the photosynthetic and respirational carbon fluxes, i.e., the Net Ecosystem Productivity (NEP)? b) What is the relative role of NHL in global terrestrial ecosystem NEP? and c) What is the magnitude of the model uncertainties related to the answers to the first two questions?

## 2 Materials and Methods

We used NEP at both global and NHL (poleward of 50 °N) scales from existing CMIP6 outputs in this study. For diagnosing purposes, we also analyzed the Net Primary Productivity (NPP) and Heterotrophic Respiration (RH), since they represent the two primary components of NEP: net plant carbon uptake and respirational carbon loss due to microbial decomposition, as NEP=NPP-RH. These model outputs were obtained from Earth System Grid Federation (ESGF) (https://esgf-node.llnl.gov/search/cmip6/, accessed on Oct. 1st, 2021) which unified the standardization to provide data access to various model outputs. Each model in CMIP6 was conducted with an ensemble of simulations with different initial conditions which were categorized and labeled with four variant indices: the realization index ($r$), the initialization index ($i$),

the physics index (*p*) and the forcing index (*f*) (Eyring et al., 2016; Petrie et al., 2020). To uniformly control the model conditions in case of unexpected uncertainties, we confined the selection of model outputs to experiments with all variant indices labeled with '*1*', i.e. '*r1i1p1f1*', for consistency. In particular, the ScenarioMIP experiments endorsed a set of future global change scenarios, i.e. the combinations of SSPs and RCPs, to represent the alternative evolutions of societal development, emissions and concentrations (O'Neill et al., 2016). The RCPs are a set of four future greenhouse gas emission pathways in which the end-of-century radiative forcing approaches four target levels (2.6, 4.5, 6.0, and 8.5 $W/m^2$), i.e. RCP2.6, RCP4.5, RCP3.7, RCP8.5 (van Vuuren et al., 2011). The four target forcing levels are set to be realized by altering future greenhouse gas emissions and by changing underlying socioeconomic projections. The SSPs were developed to describe a set of five future global socio-economic development scenarios (SSP1 to SSP5). Four future scenarios with different SSP and RCP combinations in this study, including SSP1+RCP2.6 (SSP126), SSP2+RCP4.5 (SSP245), SSP3+RCP7.0 (SSP370), SSP5+RCP 8.5 (SSP585), were considered in this study to cover a variety of future climate change projections. Overall, ten models were selected in this study, i.e. the ACCESS-ESM1-5 (Ziehn et al., 2020), BCC_CSM2-MR (Wu et al., 2019), CanESM5 (Swart et al., 2019), NorESM2-LM (Seland et al., 2020), NorESM2-MM (Seland et al., 2020), CESM2-WACCM (Gettelman et al., 2019; Lawrence et al., 2019), CMCC-CM2-SR5 (Cherchi et al., 2019), EC-Earth3-Veg (Wyser et al., 2020), IPSL-CM6A-LR (Dufresne et al., 2013), MPI-ESM1-2-LR (Mauritsen et al., 2019; Reick et al., 2013). The NorESM2-LM and NorESM2-MM share the same horizontal resolution of ocean and sea ice but differ in the horizontal resolution of land and atmosphere and varies in some parameter settings in the atmosphere component. The detailed information with land and atmosphere components and spatial resolutions, as well as key relevant model features are listed in Table 1.

We used monthly NEP, NPP, RH, 2-m air temperature (TAS) and atmospheric $CO_2$ concentration from the ten CMIP6 models over the historical period (1980-2014) and the four future scenarios (2015-2100) in our analyses. Area-weighted sum of NEP, NPP, RH and NBP, as well as area-weighted mean of TAS from different models and scenarios at global and NHL scales were calculated. Non-land fractions of grid cells were excluded in the calculation. The bottom layer (i.e., the layer nearest to the land surface) atmospheric $CO_2$ concentration was aggregated into global and NHL scales, too. Note that only four out of the ten models have available $CO_2$ data to date. The calculated monthly values from original outputs were further aggregated into the yearly scale for analysis. The annual model outputs with various spatial resolutions were resampled based

on the model grids of BCC-CSM2-MR with a grid resolution of around 1 degree (mesh size: 320 ×160) for generating the spatial trend maps. The ensemble model projections and uncertainties of NEP, NPP, RH, and TAS were evaluated by calculating the multi-model mean (μ) and standard deviations (SD, σ) of the yearly model outputs at both the global and NHL scales. Meanwhile, the contribution of model SD relative to the mean is quantified by the coefficient of variation (CV; CV = σ/μ) to interpret the relative model uncertainty. We estimated the temporal trends of μ and SD using linear least square regression method to quantitatively illustrate the ensembled model behavior against time. Additionally, the sensitivity analyses were performed by calculating the relative changes in carbon fluxes to their current levels (represented by the mean of 2010-2015) in response to the temperature rises at an increment of 1°C (Pg C/°C) or atmosphere CO2 concentration at an increment of 1 ppm (Pg C/ppm) for each model at both the global and NHL scales. Finally, we evaluated trends of the NHL carbon fluxes changes relative to the global carbon fluxes changes under the future scenarios. The flux changes were calculated using the future annual carbon fluxes subtract the 2015 carbon flux.

For the purpose of better understand the uncertainties of CMIP6 future projections, we used the land carbon budget from the Global Carbon Project (GCP; Friedlingstein et al., 2020) to benchmark the CMIP6 estimates in the historical period (1980-2014), although this is not the main purpose of this study. Such comparison would be useful because it can infer the potential biases in CMIP6 projections if we consider GCP data as the most reliable estimates of the historical carbon budget. However, only seven out of the ten CMIP6 models output the Net Biosphere Productivity (NBP), which is the difference between NEP and disturbance-induced carbon loss (e.g., fire emissions) and land use change emissions. In addition, many models and GCP data don't provide the disturbance and land use change emissions separately, making it challenging to conduct a detailed comparison between the two data sources at the detailed level. Meanwhile, only global carbon budget was provided by GCP. Thus, we only compare NBP using the available data at the global scale.

## 3 Results

### 3.1 Magnitudes of global and NHL NEP and NBP

Figure 1 shows the annual NEP in the historical (1980-2014) and future (2015-2100) periods under the four global change scenarios from the ten CMIP6 models. On average, the CMIP6 models indicate a strong global terrestrial ecosystem NEP of

4.48 ± 0.54 Pg C/year (annual mean ± interannual standard deviation) during the historical period, with a large spread across

individual models (Figure S1). Meanwhile, the CMIP6 models suggest the global NBP of the seven available models (Figure

S2) as 0.99 ± 0.68 Pg C/year. As a reference, the estimates from the GCP show the global terrestrial ecosystems as a consistent

carbon sink during the historical period at 2.43 ± 0.97 Pg C/year, which is about half lower than the model ensemble mean

NEP but higher than the NBP estimates. The models also estimate positive NHL NEPs as 0.56 ± 0.11 Pg C/year during the

historical period.

Over the future years, the CMIP6 models generally suggest positive NEP over the global terrestrial ecosystems under all

four scenarios (5.56 ± 0.88, 6.69 ± 0.78, 7.26 ± 0.98 and 8.13 ± 1.56 Pg C/year for SSP126, SSP245, SSP370 and SSP585,

respectively according to the mean of the ten models). For NHL, the NEP is estimated as 0.79 ± 0.59, 0.95 ± 0.14, 0.94 ± 0.16

and 1.01 ± 0.18 Pg C/year for SSP126, SSP245, SSP370 and SSP585, respectively. However, a few models indeed suggest

the global terrestrial ecosystems with negative NEP at the end of the 21st century under SSP126, such as CanESM5 and EC-

Earth3-Veg. In the NHL, while most models suggest a positive NEP, BCC-CSM2-MR estimates a carbon source even though

it shows the global ecosystem with a positive NEP, irrespective of the model scenarios.

### 3.2 Trends of global and NHL carbon fluxes in the 21st century

Relative to the average condition in 2015-2020, the CMIP6 models in average suggest the global mean TAS will increase

by 1.16, 2.45, 4.05, and 5.25 °C by the end of 21st century (2095-2100) under SSP126, SSP245, SSP370 and SSP585,

respectively. The growth of TAS in NHL is projected to increase by 2.36, 4.41, 7.08, and 9.36 °C by the end of this century

under the four scenarios respectively, which are exclusively higher than the global levels (Figure S3 and Table 2). The

atmospheric $CO_2$ concentrations are projected to increase at similar rates during 2015-2100 at global and NHL scales at 0.52,

2.36, 5.43 and 8.51 ppm/year under SSP126, SSP245, SSP370 and SSP585, respectively (Figure S6).

In response to the elevating temperature, NPP and RH from the CMIP6 models (Figure S4 and S5) show positive trends

under all four scenarios and the trends are larger under the warmer scenarios at both global and NHL scales. Global NPP will

increase at rates of 65.72, 196.48, 294.87 and 387.75 Tg C/year$^2$ under SSP126, SSP245, SSP370 and SSP 585, respectively.

NHL NPP are projected to grow at rates of 16.16, 41.33, 61.06, and 79.32 Tg C/year$^2$ accordingly. Except SSP126, similarly

positive but generally smaller trends were found for RH at global scales (Figure S5, Table 2) with the rates of 87.15, 173.39,

254.43 and 318.31 Tg C/year$^2$ under the four scenarios. The NHL RH trends are 18.64, 36.27, 55.39 and 72.56 Tg C/year$^2$. Normalized by the area, the growth rates are 0.44, 1.33, 1.99 and 2.62 g C/year$^2$ for global NPP over the four scenarios respectively. The area-normalized growth rates in the NHL NPP are 0.54, 1.37, 2.03 and 2.63 g C/year$^2$, respectively. Area-normalized global RH growth rates are 0.59, 1.17, 1.72 and 2.15 g C/year$^2$ while the area-normalized NHL RH growth rates are 0.62, 1.20, 1.84 and 2.41 g C/year$^2$ under the four scenarios, respectively. These results indicate that a faster average

growing NPP and RH in the NHL than the global average. The fast-growing RH cancels a large part of the NPP growth and resulted in small growing NEPs.

        CMIP6 models show a trend of NEP that first increases until the middle of the 21st century and then decreases at both NHL and global scales under SSP126. Overall, they show a slightly decreasing trend at NHL (-2.84 Tg C/year$^2$) and global (-22.50 Tg C/year$^2$) scales during 2015-2100 under SSP126. The trends are positive under SSP245 at 8.93 Tg C/year$^2$ at the

global scale, and 2.54 Tg C/year$^2$ for NHL. Under SSP370 and SSP585, the positive trends become more prominent: they are 20.08 and 44.40 Tg C/year$^2$ at the global scale, and 3.08 and 4.27 Tg C/year$^2$ in the NHL under SSP370 and SSP585, respectively.

### 3.3 Divergent carbon flux estimations among the CMIP6 models

        Large uncertainties of estimated global and NHL NEP were found, measured by the standard deviation (SD) across the

CMIP6 models. The average SD for global NEP over the historical period was 2.85 PgC/year, and it will expand to 3.96, 4.51, 5.44 and 5.60 Pg C/year by the end of the 21$^{st}$ century under SSP126, SSP245, SSP370 and SSP585, respectively. Specifically, the model uncertainties of global and NHL NEP conserve under SSP126 with small shrinking trends of SD values (-2.84 Tg C/year$^2$ and -0.22 Tg C/year$^2$ for global and NHL respectively; Table 2). For SSP245, SSP370 and SSP585, the model uncertainties tend to expand towards the end of this century for both global and NHL scales. The model uncertainties are the

largest under SSP370 and SSP585. Globally, the mean NEP values for SSP370 and SSP585 are 6.08 Pg C/year and 7.77 Pg C/year, respectively, during the (2095-2100) with concomitant large SDs of 7.84 Pg C/year (CV = 129%) and 8.53 Pg C/year (CV = 109.78%). It is worth noting that the mean NEP values for SSP370 and SSP585 in NHL are 0.77 Pg C/year and 0.84 Pg C/year, respectively, during 2095-2100, while the SDs are relatively large: 1.64 Pg C/year (CV = 213.00%) and 1.86 Pg C/year (CV = 221.43%) accordingly. Similarly, large uncertainties for NPP and RH were identified. The average SD for global

and NHL NPP over the historical period were 14.89 and 1.51 PgC/year, respectively, and they are projected to expand at rates of 50.10, 138.01, 219.68, 284.02 TgC/year (global) and 4.64, 8.87, 18.07 and 26.87 TgC/year (NHL) under SSP126, SSP245, SSP370 and SSP585, respectively. For RH, the global and NHL average SD over the historical period were 16.15 and 1.66 PgC/year, respectively, and they are projected to expand at rates of 18.54, 36.27, 55.39, 72.56 TgC/year (global) and 4.06, 7.76, 16.63, and 23.52 TgC/year (NHL) under SSP126, SSP245, SSP370 and SSP585, respectively.

The large uncertainties of NEP are likely due to the uncertain responses of NPP and RH to the temperature changes and $CO_2$ fertilization effects in each model. The SDs of TAS projections by the end of the 21[st] century are 2.52, 2.79, 2.68, 2.71 °C in the NHL, which are much larger than those of global TAS at 0.83, 0.84, 1.04, and 1.27 °C, under SSP126, SSP245, SSP370 and SSP585, respectively. As shown in Figure 2, the CMIP6 estimated annual carbon fluxes have strong linear relationships to TAS. For NPP, a 1 °C increase of global TAS corresponds to an increase of global annual NPP from 0.47 to 13.34 Pg C/year; in the NHL, the range spans from 0.28 to 0.95 Pg C/year. Global annual RH will increase at rates from 1.06 to 11.12 Pg C per 1 °C increase of global TAS, and the rates are between 0.28 and 1.29 Pg C/year for the NHL annual RH. All the lowest sensitivities are estimated by ACCESS-ESM-1-5 and the highest sensitivities are from CanESM5. As the residual of NPP and RH, the sensitivities of NEP to TAS are more complicated: the global annual NEP will change at a rate between -0.59 (by ACCESS-ESM-1-5) and 2.21 Pg C/year (by CanESM5) per 1 °C increase of global TAS; and the changing rates are between -0.37 (by BCC-CSM2-MR) and 0.23 Pg C/year (by CanESM5) for the NHL annual NEP. The 'carbon fluxes vs. $CO_2$ concentration' and 'carbon fluxes vs. temperature rise' demonstrate similar linear relationships, as shown in Figure 3. The global NPP increases at a rate from 0.037 PgC/year per ppm $CO_2$ concentration rise by IPSL-CM6A-LR to 0.064 PgC/year by BCC-CSM2-MR, and NHL NPP increases at a rate from 0.008 PgC/year by MPI-ESM1-2-LR to 0.011 PgC/year by BCC-CSM2-MR. The global RH increases at a rate from 0.030 PgC/year per ppm $CO_2$ concentration rise by IPSL-CM6A-LR to 0.058 PgC/year by BCC-CSM2-MR globally, and from 0.007 PgC/year by IPSL-CM6A-LR to 0.015 PgC/year by BCC-CSM2-MR at the NHL scale. The NEP show contrasting trends at the two different spatial scales relative to the $CO_2$ concentration rise by BCC-CSM2-MR: at the global scale, NEP is positive correlated with $CO_2$ concentration, while at the NHL scale they are negatively correlated. The other three models show slightly positive trends of NEP fluxes relative to the $CO_2$ concentration rise at both scales. There remains a strong linear relationship between TAS and atmospheric $CO_2$

concentrations irrespective of the model scenarios (Figure S7), which could explain the similar trend patterns of carbon fluxes change in response to the TAS and $CO_2$ concentration rise in Figure 2 and Figure 3.

### 3.4 Latitudinal distributions of NEP

Figure 4 shows average NEP in the 10°-latitudinal bins between 60°S and 90°N in the historical, the early (2015-2024), the middle (2050-2059) and the end (2091-2100) decades of the 21$^{st}$ century under the four scenarios. Overall, the global ecosystems are projected as a stronger carbon sink under SSP245, SSP370 and SSP585 than the historical period for most of the latitudes except the polar region (>80 °N) where the NEP remains relatively constant. Under SSP126, there is a drawdown during 2091-2100 between 20 °S to 10 °N. Among all the latitudinal bins, the tropical regions near the equator act as the largest carbon sink with the highest uncertainties. However, the uncertainties at 60 °N and 70 °N are exclusively larger relative to the absolute values of NEP in this region (i.e., the CV values), which are 109.44%, 264.11% under SSP126, 86.37%, 173.89% under SSP245, 106.92%, 364.27% under SSP370, and 119.60%, 484.50% under SSP585, comparing with those near the equator of 100.32%, 58.94%, 80.46%, 54.58% for the four future scenarios, accordingly.

### 3.5 Spatial pattern of trends of NHL carbon fluxes

According to the average of CMIP6 models, Figure 5 shows significant positive trends of NPP and RH, but mixed trends of NEP in the NHL under all of the four scenarios. With growing radiative forcing or temperature from SSP126 to SSP585, the positive trends of NPP and RH increase everywhere in the NHL. The spatial pattern of NEP trends is more complicated. Under SSP126, most of the forested area in the NHL are projected to have significantly decreasing NEP, while the other regions show no significant trends. More area turns to have significantly positive and larger NEP trends from SSP126 to SSP245 and SSP370 in response to larger radiative forcing levels. Under SSP585 which shows the highest level of radiative forcing and global warming, most of the NHL NEP, particularly areas covered by forest, are projected to have significant positive trends, while the NEP in the tundra area of Northern Canada and Siberia are in contrast have significant negative trends.

### 3.6. The role of NHL in future global carbon fluxes changes

The CMIP6 models show consistent positive contribution of the NHL to the global carbon fluxes changes since 2015, measured by slopes of linear regression models between the NHL and global numbers (Figure 6). On average, the CMIP6 models estimate that NHL contributes 16% of global NPP increase under SSP126 and 20% under the other three scenarios and contributes 23%-26% of global RH increase under the four scenarios. For NEP, the NHL's contributions are between 7% and 11%. However, it is worth noting that some of these contributions are with high uncertainties from different models. For

example, CanESM5 generally projects largest increases of global and NHL NPP and RH but stands out to suggest the lowest NHL contribution (i.e., the smallest slopes) to global NPP and RH. The uncertainties (measured by the standard deviation of the slopes estimated by the ten models) are relatively lower for NPP and RH and scenarios with lower radiative forcing levels but become high for NEP under high radiative forcing scenarios. For instance, the uncertainties could be as high as five-fold of the contribution estimated by the multi-model means for NEP under SSP370 and SSP585.

### 4 Discussion

In this analysis, we presented the quantification of the future magnitudes, trends, patterns and uncertainties of terrestrial ecosystem carbon fluxes from an ensemble of ten CMIP6 models, with a particular focus on the Arctic-Boreal regions in the Northern high latitudes. The CMIP6 models estimate the global terrestrial ecosystems as a strong carbon sink but with a magnitude that is 2.06 Pg/year or 85% higher than the estimates from the benchmarking global carbon project, suggesting

consideration of bias corrections when using CMIP6 modeled carbon fluxes for other applications, particularly those sensitive to the magnitude of these carbon fluxes.

On average, the CMIP6 models project large increases of NPP and RH in the global and NHL terrestrial ecosystems in the future, while the NHL is projected to grow 1.43, 1.13, 1.31, 1.40 times faster for NPP and 1.47, 1.46, 1.58, 1.55 times faster for RH, under SSP126, SSP245 SSP370 and SSP585 than those at the global scale (Table 2). This is because of the faster

increase of temperature, larger $CO_2$ fertilization effect, and higher sensitivities to the warming climate (Figure 2) in the NHL. Such concurrent rising NPP and RH was widely evidenced and discussed in previous literature. Jeong et al., (2018) showed that long-term measurements of $CO_2$ revealed increasing carbon cycling rates and decreasing soil carbon residence time in the Arctic. On one hand, greening of the world was widely identified due to more favorable vegetation growth conditions promoted

by a warming climate (Piao et al., 2020; Zhu et al., 2016), and warmer temperature and $CO_2$ fertilization was revealed to enhance the terrestrial gross primary production in the NHL (Liang et al., 2018; Myers-Smith et al., 2020; Wenzel et al., 2016). On the other hand, the increases in RH in response to temperature rise could be attributed to two major reasons (Bond-Lamberty et al., 2018). One reason for the rising RH could result from more active soil bacteria metabolism, and thus enhanced SOM mineralization due to rising temperature (Crowther et al., 2016; Lu et al., 2013). The second reason could be the more abundant availability of substrates for metabolism from accelerated ecosystem carbon uptake and debris production (Bond-Lamberty et al., 2018).

The terrestrial ecosystem carbon cycle is complex and many past and ongoing ecological studies sought to understand the underlying mechanisms. Long-term measurements at FLUXNET sites have evidenced greater bioavailable carbon stock due to the faster increasing gross primary production than the concurrent rises of ecosystem respiration in response to the climate change (Falge et al., 2002). However, contradictory conclusions were drawn in some regions of the world where reduced soil carbon stocks were found due to more carbon efflux than influx (Naidu and Bagchi, 2021). The case in the NHL is even more special, partly because the biological processes such as the vegetation phenology and soil decomposition are especially sensitive to climate change due to the extremely cold environment and the relatively faster temperature change rates (McGuire et al., 2009; Richardson et al., 2018). The thawing of permafrost is changing the soil water balance and increasing the thickness of the active layer, which renders the ancient carbon under potential decomposition (Belshe et al., 2012; Schuur et al., 2015; Schuur & Abbott, 2011). Moreover, the terrestrial carbon fluxes are influenced by the evolutions of various other climate factors, such as precipitation, soil moisture and atmospheric nitrogen deposition (Naidu and Bagchi, 2021; Sierra et al., 2015; Yue et al., 2017). Besides the disturbance-induced carbon loss, the carbon balance in the terrestrial ecosystems will be determined by the difference between rising primary productivity and the accelerated soil carbon decomposition driven by the interplay of multiple climate drivers (McKane et al., 1997; Sistla et al., 2013). These complex processes have been reflected in the results of our CMIP6 analysis. As the residual between the carbon influx (NPP) and efflux (RH), global and NHL NEP are projected to have more complicated changing patterns. The global and NHL NEP are growingly positive in the future, but at lower rates than NPP and RH. While global NEP is generally higher under warmer scenarios, NHL NEP will be at similar levels by the end of the 21st century under different warming levels (e.g., SSP245, SSP370, SSP585; Figure 1). This is partially

due to the varying response of different ecosystems to the warming climate, as forest-dominated area is becoming a larger

carbon sink and tundra-dominated area is likely becoming a stronger carbon source (Figure 4).

Yet, it is important to note that there remain large uncertainties of the magnitudes and trends of the carbon balance in the global and NHL terrestrial ecosystems. The underlying carbon cycling processes are difficult to quantify and are poorly constrained in current ESMs (Bradford et al., 2016). Sensitivities of carbon fluxes in ESMs are divergent in responses to different climate change drivers (e.g., Figure 2 and Figure 3), such that model uncertainties are pronounced in various aspects

(Bradford et al., 2016). Although different land surface models share the similar carbon fluxes transfer mechanisms among different carbon pools, they are diversified in the pool structures (Shao et al., 2013; Yan et al., 2014) and parameterizations (Luo and Schuur, 2020). For example, CMIP6 models were found to inaccurately estimate Leaf Area Index (LAI), an essential biophysical variable that drives carbon cycle and many other ecological processes (Song et al., 2021). Song et al. (2021 )suggested that most CMIP6 models were not able to correctly reproduce the magnitudes short-to-long-term temporal

variability of LAI, although they showed improvement in estimating seasonal LAI variations compared with CMIP5 models. While it is hard to distinguish how the improvements of LAI estimation in CMIP6 models might contribute to their performance on estimating carbon fluxes, their physical linkages are clear. Any underestimation of LAI usually leads to lower NPP estimations and possibly higher RH due to the lower LAI-induced weaker cooling effects on the soil and therefore may result in lower NEP. Better seasonal variation of LAI may indicate better capture of the growing season length of vegetation and the

annual carbon budget (Piao et al., 2019). In addition, the categorizations of plant functional types (PFTs) are also different among the 10 ESMs (Table 1), for example, CanESM5 has 9 PFTs while CESM2-WACCM has 15 PFTs plus additional crop types. Most models have the nitrogen cycles coupled with carbon cycles with exceptions of CanESM5 and IPSL-CM6A-LR (Table 1). For compensating the effects of nutrient limitation, IPSL-CM6A-LR adopts the 'downregulation' function to limit the maximum photosynthesis rates to account for nutrient limitations (Boucher et al., 2020), while the CanESM5 has no

nutrient limitations accounted (Swart et al., 2019). This could be one of the reasons CanESM5 has the largest sensitivities of NPP and RH fluxes in response to the climate change (Figure 2). Comprehensive and standard validations of multiple variables are needed to assess the model performance and uncertainties of biogeochemical simulations across CMIP6 models (Spafford and MacDougall, 2021).

In our analysis, the uncertainties of the carbon fluxes across the CMIP6 models tend to increase over time, and they grow faster under warmer scenarios. The NHL NEP has more relative uncertainties as opposed to the mean compared with global NEP, and this difference is more pronounced in scenarios with higher radiative forcing levels. By 2100, the CMIP6 models suggest the NHL as a carbon sink of 0.54 ±0.77, 1.01 ±0 .98, 0.97 ±1.62, and 1.05 ±1.83 Pg C/year under SSP126, SSP245, SSP370 and SSP585, respectively, which are exclusively larger than the previous C4MIP results under IPCC SRES A2 scenario with temperature rise of approximately 3.4 (2.0–5.4) °C by 2100 (0.3 ± 0.3 PgC/ year; Qian et al., 2010). The relative uncertainties (SD/mean) for the four scenarios are 143.59%, 97.03%, 167.01% and 174.29% which are at the similar or larger levels than the C4MIP results (100%), indicating the uncertainty level is not reduced in the new models. Moreover, models show distinct sensitivities of carbon fluxes in response to the future temperature rise. While NPP and RH show uniformly positive response to temperature rise, NEP changes could be either positive or negative for different models. The uncertainties in soil carbon dynamics and various projections of soil carbon stock and changes in different CMIP5 models were broadly evaluated and discussed in previous studies (Friedlingstein et al., 2014; Todd-Brown et al., 2013, 2014; Yan et al., 2014). Recent evaluations of soil carbon stock and sequestration of CMIP6-LUMIP models also showed large differences among different CMIP6 models, which in another way indicates the possible uncertainties of soil carbon dynamics stemming from simulating the land-use impacts in different CMIP6 models (Ito et al., 2020). All the CMIP6 model results present in this analysis do predict rising NPP and RH in response to temperature rise in the future but with divergent trends and patterns. Consequently, large uncertain or even irreconcilable NEP results in the NHL is shown among different models.

## 5 Conclusion

The climate model intercomparison project is a major approach to quantify and understand the future terrestrial ecosystem carbon cycle and its interactions with the climate system. In this study, we presented the trends and patterns of future projections of carbon fluxes (particularly the net ecosystem productivity) in the global and northern high-latitude ecosystems, from a set of the most up to date CMIP6 models. Based on the average of the CMIP6 models, our analysis showed that global and NHL ecosystems were and would continue to be carbon sinks, although large uncertainties were found for the size and trends of the carbon sinks among different CMIP6 models, which are not obviously attenuated compared with previous model

intercomparison project results. Although the warming levels and sensitivity of ecosystems to the warming temperature are

higher in the NHL, the contribution of NHL to the global NEP increase is small, however with larger relative uncertainties.

The model uncertainties are pronounced in the historical simulations and are projected to expand wider in the future under

scenarios with larger radiative forcing levels. These results revealed the emergent necessity to make endeavors to bridge the

knowledge gaps between process parameterization and representations of various ESMs and the real-world processes, as well

as to deepen the understanding of the underlying mechanisms of the feedforward and feedback roles of the NHL ecosystem in

response to climate change.

## Data and code availability

The CMIP6 model results are public available at ESGF website: https://esgf-node.llnl.gov/search/cmip6/. The codes for processing the data and generating the figures are available at: https://github.com/qhgogogo/CMIP6-carbonflux.

## Author contributions

Han Qiu and Min Chen designed the study, processed the data and prepared the original manuscript. Dalei Hao, Yelu Zeng and Xuesong Zhang contributed subsequent analysis, and helped writing and editing the manuscript.

## Competing interests

The contact author has declared that neither they nor their co-authors have any competing interests.

## Acknowledgments

We acknowledge the World Climate Research Program, which, through its Working Group on Coupled Modelling, coordinated and promoted CMIP6. We thank the climate modeling groups for producing and making available their model
output, the Earth System Grid Federation (ESGF) for archiving the data and providing access, and the multiple funding agencies who support CMIP6 and ESGF, as well as the efforts of all involved modeling centers.

Han Qiu, Yelu Zeng and Min Chen are supported by the National Aeronautics and Space Administration (NASA) through Terrestrial Ecology: Arctic Boreal Vulnerability Experiment (ABoVE) grants NNH18ZDA001N (award number 80HQTR19T0055) to Min Chen. Xuesong Zhang acknowledges support from the U.S. Department of Agriculture, Agricultural
Research Service. USDA is an equal opportunity provider and employer.

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

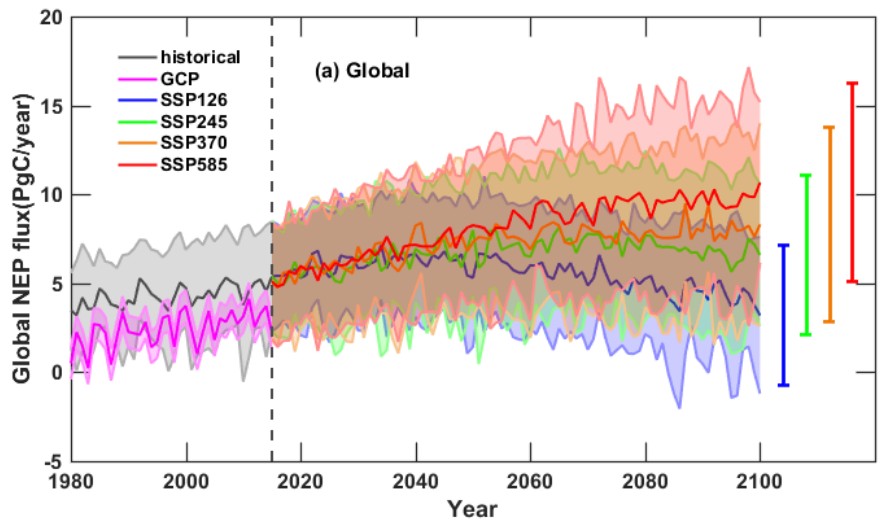

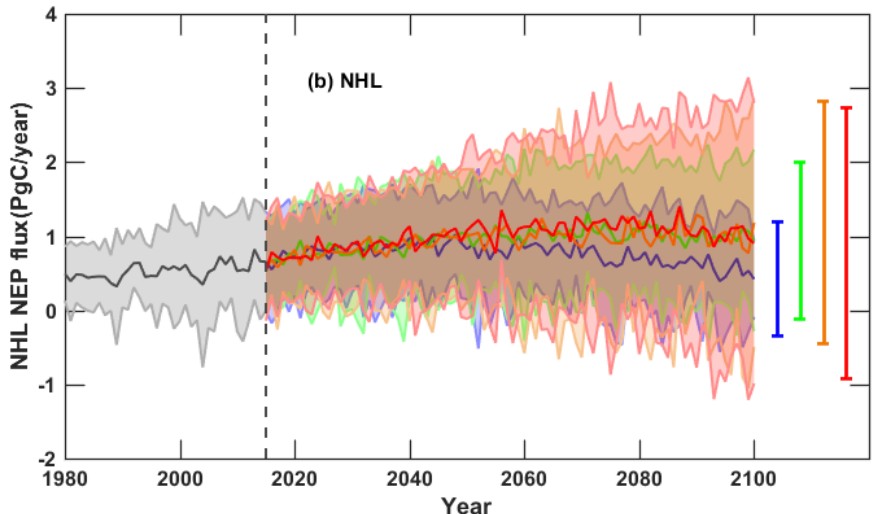

**Figure 1: The annual mean and SD of NEP of the ten CMIP6 models during the historical period (1980-2014) and the future period (2015 - 2100) under four global change scenarios at the global (a) and Northern High Latitude (NHL) (b) scales. The shaded area indicates the SD values across the models. Error bars at the right of the panels show the mean SD of NEPs during 2095-2100 under each of the four scenarios.**

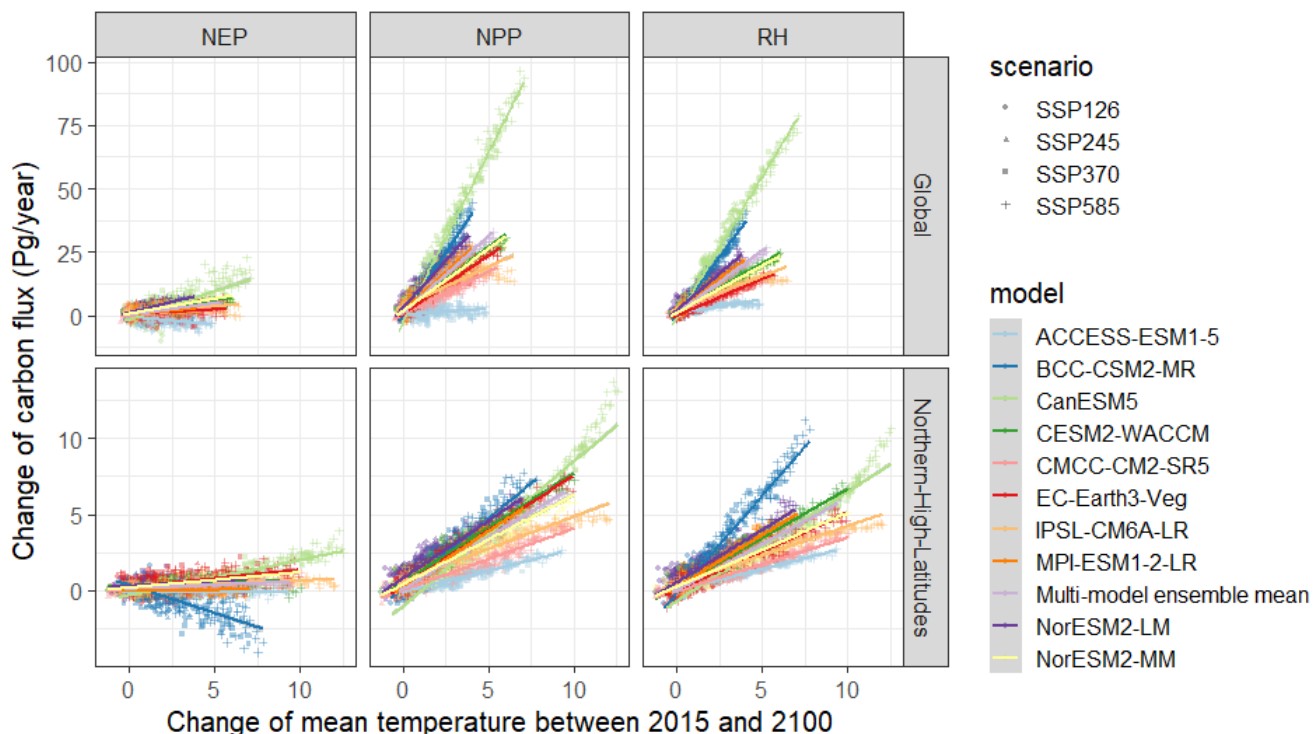

**Figure 2: Sensitivity of carbon fluxes changes in response to the TAS changes (relative to the 2015 values) at global and NHL scales for each CMIP6 model under the four future scenarios.**

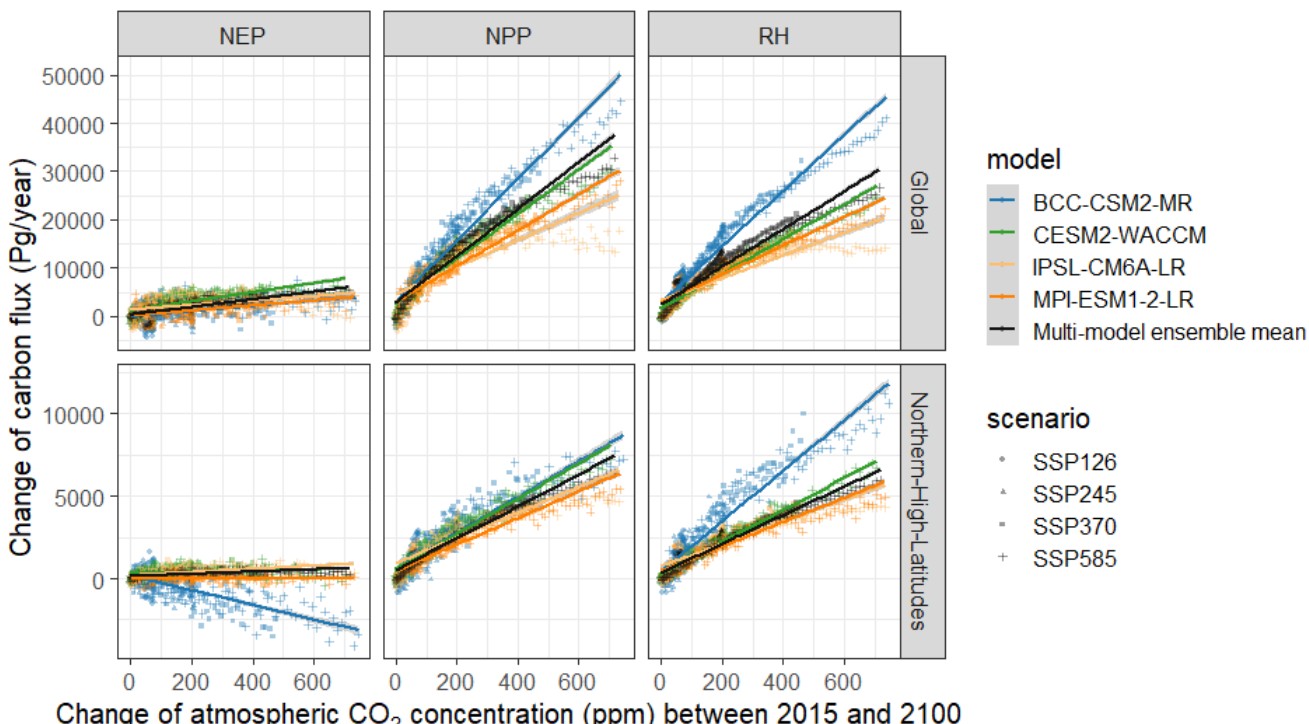

**Figure 3: Sensitivity of carbon fluxes changes in response to the CO₂ concentration changes (relative to the 2015 values) at global and NHL scales for each CMIP6 model under the four future scenarios. Only available data from four CMIP6 models was used for producing this figure.**

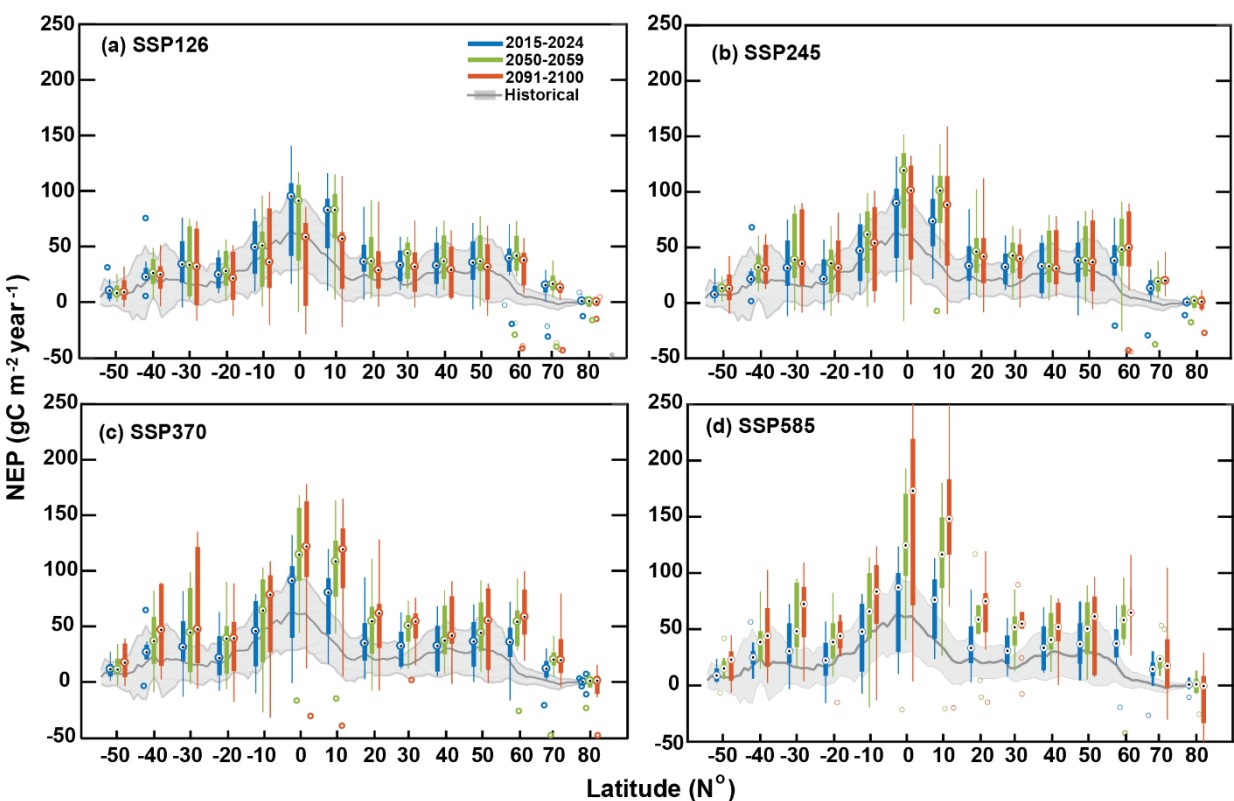

**Figure 4: Latitudinal distributions of NEP in the historical period and under different future scenarios. The grey lines with bands are the historical multi-model mean and uncertainties of NEP. The boxplots are the future NEP distributed in each 10° bin between 60°S and 90°N under: (a) SSP126, (b) SSP245, (c) SSP370, (d) SSP585, during the early (2015-2024), the middle (2050-2059) and the end (2091-2100) decades of the 21st century.**

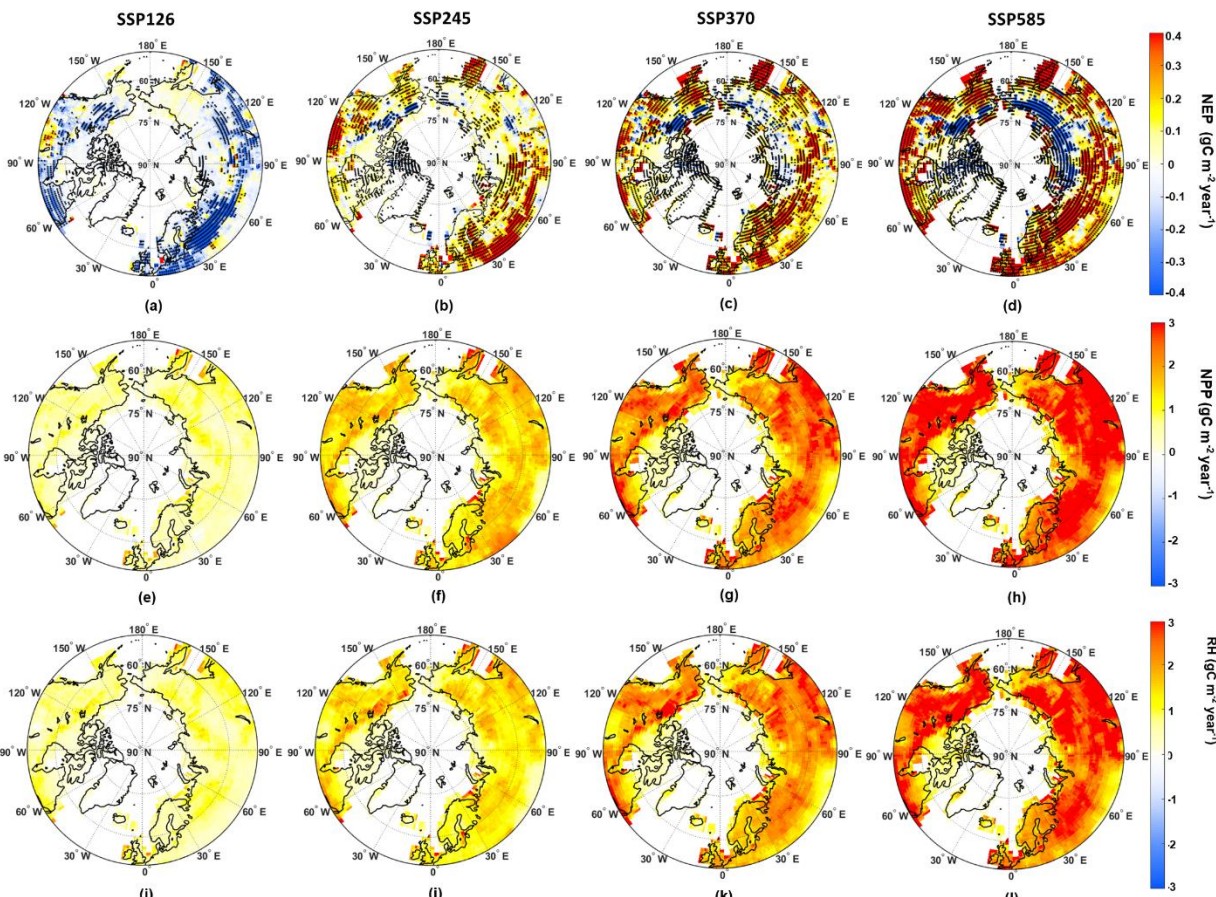

**Figure 5: The spatial distributions of the trends of NHL carbon fluxes under different future scenarios. The rows of the panels are NEP, NPP and RH from top to bottom and the columns of the panels are SSP126, SSP245, SSP370 and SSP585 from left to right. Unit is g C m$^{-2}$ year$^{-1}$. The black dots on the NEP maps denote significance of the regression values (p<0.05) when fitting the carbon fluxes trends within each grid. Most of the model grids show significance of the regression for NPP and RH and are not shown on the maps.**

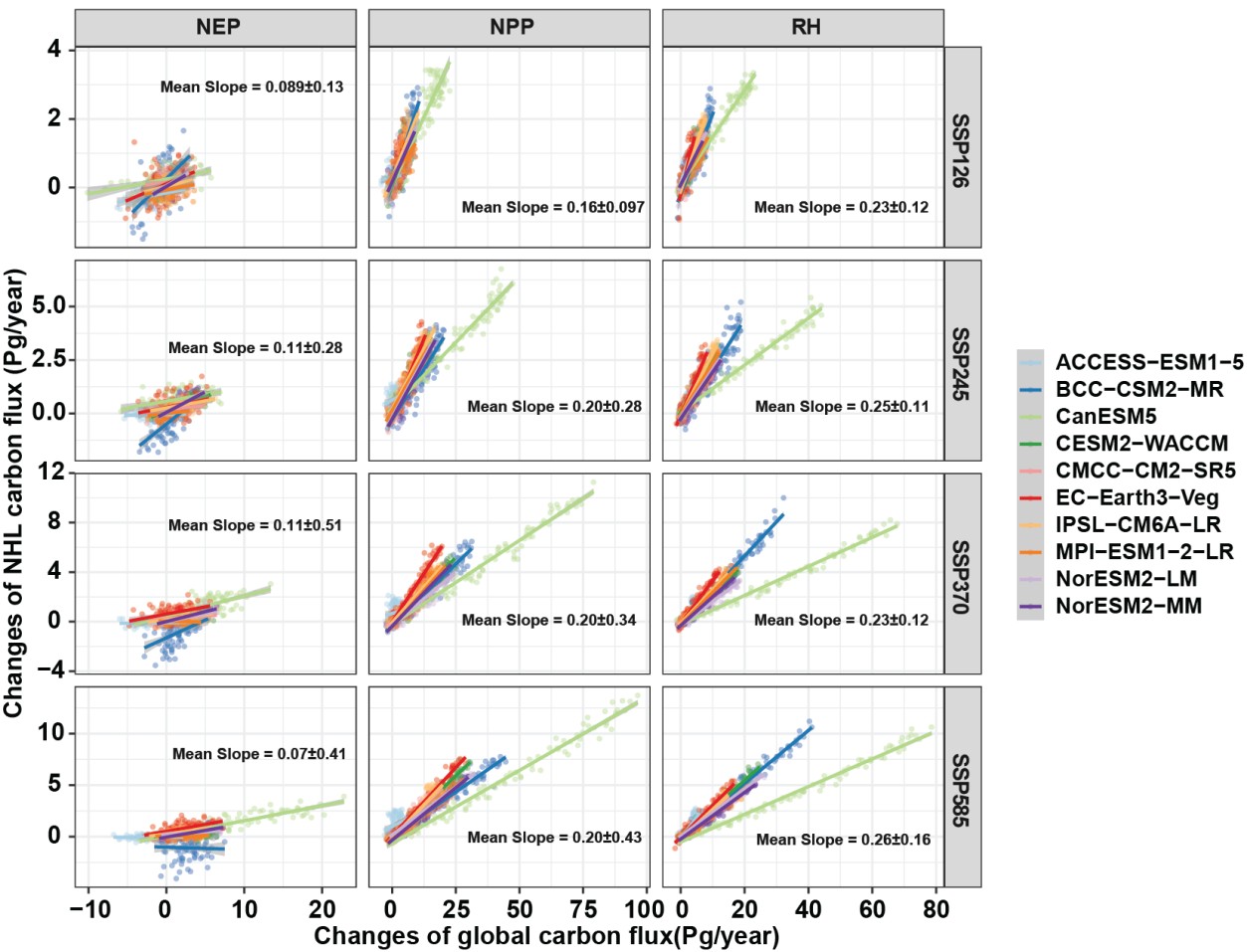

Figure 6: Changes of NHL carbon fluxes relative to the changes of global carbon fluxes, as indicated by the ten CMIP6 models.

**Table 1: The CMIP6 models analyzed in this study, the model land and atmosphere components, spatial resolutions and key relevant model features are listed.**

| Models | Component models (longitude×latitude grids) | | Soil layers | N cycle | Number of Plant function types (PFTs) | Dynamic vegetation | CO₂ fertilization effect |
| | Atmosphere model | Land component model | | | | | |
|---|---|---|---|---|---|---|---|
| **ACCESS-ESM1-5** | HadGAM2 (192×145) | CABLE2.4 (192×145) | 6 | Yes | 13 | No | Yes |
| **BCC-CSM2-MR** | BCC_AGCM3_MR (320×160) | BCC_AVIM2 (320×160) | 10 | Yes | 15 | Yes | Yes |
| **CanESM5** | CANAM5 (128×64 ) | CLASS3.6/CTEM1.2(128×64) | 3 | No | 9 | Yes | Yes |
| **NorESM2-LM*** | CAM-OSLO (144×96) | CLM5 (144×96) | 15 | Yes | 15+ crop PFTs | Yes | Yes |
| **NorESM2-MM*** | CAM-OSLO (288×192) | CLM5 (288×192) | 15 | Yes | 15+ crop PFTs | Yes | Yes |
| **CESM2-WACCM** | WACCM6 (288×192) | CLM5 (288×192) | 15 | Yes | 15+ crop PFTs | Yes | Yes |
| **CMCC-CM2-SR5** | CAM5.3(288×192) | CLM4.5, BGC mode (288×192) | 15 | Yes | 15+ crop PFTs | No | Yes |
| **EC-Earth3-Veg** | IFS cy36r4 (512×256) | HTESSEL (land surface scheme built in IFS) and LPJ-GUESS v4 (512×256) | 2 | Yes | 11 | Yes | Yes |
| **IPSL-CM6A-LR** | LMDZ (144×143) | ORCHIDEE v2.0, Water/Carbon/Energy mode (144×143) | 11 | No | 15 | No | Yes |
| **MPI-ESM1-2-LR** | ECHAM6.3 (192×96) | JSBACH3.20 (192×96) | 5 | Yes | 13 | Yes | Yes |

\* the same models but run at different spatial resolutions

**Table 2: Future trends and percent changes relative to 2010-2014 for the multi-model mean NEP, NPP, RH and TS as well as their uncertainties (SD across models) of the ten CMIP6 models.**

| Scenarios | Trends of ensembled model mean (Tg C/year$^2$ or ºC/year ; percent change relative to 2010-2014) | | | | Trends of model uncertainty (TgC/year$^2$ or ºC/year; percent change relative to 2010-2014) | | | |
|---|---|---|---|---|---|---|---|---|
| | SSP126 | SSP245 | SSP370 | SSP585 | SSP126 | SSP245 | SSP370 | SSP585 |
| **Global NEP** | -22.50 (20.0%) | 8.93 (44.5%) | 20.08 (56.8%) | 44.40 (75.6%) | -2.84 (5.0%) | 22.98 (17.7%) | 35.03 (26.4%) | 51.75 (33.5%) |
| **Global NPP** | 65.72 (9.7%) | 196.48 (15.9%) | 294.87 (20.5%) | 387.75 (24.5%) | 50.10 (23.5%) | 138.01 (38.7%) | 219.68 (53.1%) | 284.02 (63.5%) |
| **Global RH** | 87.15 (9.0%) | 173.39 (13.6%) | 254.43 (17.6%) | 318.31 (20.6%) | 68.59 (16.0%) | 136.77 (27.8%) | 197.18 (38.0%) | 228.03 (42.5%) |
| **Global TAS** | 0.013 | 0.031 | 0.050 | 0.066 | 0.0027 | 0.0033 | 0.0043 | 0.0054 |
| **NHL NEP** | -2.43 (22.8%) | 2.54 (53.5%) | 3.08 (52.4%) | 4.27 (62.9%) | -0.22 (-3.1%) | 5.37 (10.4%) | 11.04 (30.2%) | 14.03 (45.2%) |
| **NHL NPP** | 16.16 (13.9%) | 41.33 (22.4%) | 61.06 (26.9%) | 79.32 (34.3%) | 4.64 (19.3%) | 8.87 (22.9%) | 18.07 (41.8%) | 26.87 (55.5%) |
| **NHL RH** | 18.54 (13.2%) | 36.27 (19.8%) | 55.39 (27.8%) | 72.56 (31.9%) | 4.06 (9.0%) | 7.76 (15.7%) | 16.63 (30.2%) | 23.52 (40.3%) |
| **NHL TAS** | 0.026 | 0.057 | 0.09 | 0.12 | 0.015 | 0.019 | 0.017 | 0.017 |