# Peer review of "Global and Northern-High-Latitude Net Ecosystem Production in the"

_EGUsphere, 2022_

## Author Comment (AC1)

Reviewer 1

The authors investigated the future trajectories, patterns and uncertainties of northern ecosystem carbon fluxes using an ensemble of CMIP6 climate models. They found that under future warmer climates, the CMIP6 models project an overall enhanced net carbon uptake by the ecosystems, resulted from a tradeoff between NPP and RH increases. The spread of estimates across individual models is larger than that of the global average. The paper is methodologically sound, well written, and the results are nicely presented. I have a number of line comments on the manuscript, as given below. I would like to recommend its publication with the following possible revisions or clarifications.

Line comments:

Lines 113-114: It's better to use "ten models" directly, rather than "nine models with ten datasets".

Response: we used 'ten models' instead in the revised manuscript.

Line 123: I don't understand why land surface temperature, rather than 2-m temperature, is used in this analysis. When people say ecosystem response to temperature, they often refer to air temperature.

Response: Thanks for pointing out this. In the revision, we replaced the land surface temperature with 2-m air temperature in our analyses and reported the new results in the updated manuscript. We found no essential changes to the results and findings with the new variable. But it is indeed true that the use of 2-m air temperature will potentially make our report more useful.

Line 129: Do you mean the original annual outputs from models, or the annual values aggregated from original monthly outputs?

Response: We meant the annual values aggregated from original monthly outputs. We clarified this in the revised manuscript.

Line 130: If the model data are resampled to 1-degree global grids, this is supposed to be 360x180? Please clarify.

Response: The '1-degree grid' was an approximation. In practice we resampled the model data based on the model grids of BCC-CSM2-MR whose resolution is in the middle (globally 320x160). We clarified it in the revised manuscript.

Lines 145-151: The model ensemble mean of global NEP is strongly higher than the estimate by the Global Carbon Project. What's the implications for the NHL NEP and future projections of NEP changes?

Response: Thanks for pointing out that. We focused on NEP in our analyses while Global Carbon Project (GCP) reported Net Biosphere Productivity (NBP) which took disturbance-induced carbon fluxes from NEP and thus was lower. To make a meaningful comparison between CMIP6 and GCP data, we added historical CMIP6 NBP into analysis in the revision. According to the updated Fig.1a, the CMIP6 estimated lower global NBP than GCP data, and the CMIP6 NBP is closer to GCP data than NEP,

Lines 167-172: It's interesting to look at how the NHL mean NEP compares with the global mean, and whether this difference is contributed more by NPP or RH.

Response: Thanks for the suggestion. We looked into the mean carbon fluxes and added the following sentences:

"Except SSP126, similarly positive but generally smaller trends were found for RH at global scales (Figure S4, Table 2) with the rates as 87.15, 173.39, 254.43 and 318.31 Tg C/year$^2$ under the four scenarios. The NHL RH trends are 18.64, 36.27, 55.39 and 72.56 Tg C/year$^2$, normalized by the area, the growth rates are 0.44, 1.33, 1.99 and 2.62 g C/year$^2$ for global NPP over the four scenarios respectively. The area-normalized growth rates in the NHL NPP are 0.54, 1.37, 2.03 and 2.63 g C/year$^2$, respectively. Area-normalized global RH growth rates are 0.59, 1.17, 1.72 and 2.15 g C/year$^2$ while the area-normalized NHL RH growth rates are 0.62, 1.20, 1.84 and 2.41 g C/year$^2$ under the four scenarios, respectively. These results indicate that a faster average growing NPP and RH in the NHL than the global average. The fast-growing RH cancelled a large part of the NPP growth and resulted in small growing NEPs."

Figure 2 and 3: Please also include the multi-model ensemble mean result.

Response: we added multi-model ensemble mean results in the updated Figure 2 and 3.

---

## Author Comment (AC2)

Reviewer 2

The manuscript by Qiu and colleagues analyzes future projections from models included in the Coupled Model Intercomparison Project Phase 6 (CMIP6), with a focus on carbon fluxes in the northern high latitudes (NHL) compared to rest of the globe. The authors find that the CMIP6 models project terrestrial ecosystems to remain carbon sinks, with net sequestration rates increasing with global temperature changes, and that NHL ecosystems are a relatively minor fraction of the net sink but considerably more uncertain than the global mean.

Overall, this is a straightforward and relatively simple paper documenting CMIP6 output in the NHL domain. It reads somewhat like a report and contains little novelty, using familiar tools and approaches with largely unsurprising results, especially for readers following the evolution of these models and how they represent the carbon cycle. Nonetheless, the data being reported is important to the scientific community. The manuscript would benefit from additional context and discussion, grammar checks, and potentially additional analyses. It's also unclear to me if the authors are actually analyzing the net carbon sink (Net Ecosystem Productivity vs. Net Biome Productivity).

Response: we thank the reviewer for the insightful view and comments on our study. We found the comments were very helpful and had heavily revised the manuscript according to them. Please see the detailed responses below.

Specific comments

My main technical question is why the authors chose to analyze Net Ecosystem Productivity, which is the balance between photosynthesis and ecosystem respiration, versus Net Biome Productivity, which includes other fluxes such as fire and harvest disturbance and is a land model's attempt at the complete carbon cycle (it would also be helpful to include lateral fluxes if possible). I realize the participating models represent different aspects of the carbon cycle, which may make comparisons more challenging, but the manuscript essentially equates NEP to net carbon balance, which is incorrect (although the difference may be minor). It may be worth adding Supplementary analyses to address this: does the inclusion of these other fluxes change the results shown, even for a subset of the models? Furthermore, is direct comparison of CMIP6 NEP to values from the Global Carbon Project valid given these differences? Does that partially account for the large differences shown between the two in Figure 1? The authors do not provide details on how they made this comparison, so it's not possible for the reader to tell.

Response: Thank you for the comment. We acknowledge the inaccurate expressions of the 'carbon sink' in the original submission and have corrected them throughout the manuscript from the title to the conclusion. Actually, we intended to conduct our analyses on NEP in order to focus on the main components of the carbon budget. In addition, the analysis of disturbance-induced carbon fluxes is challenging because 1) not all the ten models used in this study estimate disturbance-induced carbon fluxes, 2) the definitions of disturbance are not consistent across the models, making the inter-model comparison challenging, and 3) land-use flux is commonly included in the disturbance flux in most models, but it is an independent flux variable in GCP.

However, we agree that supplementary analyses with the disturbance-induced carbon flux would be useful. Therefore, we reported the NBP (=NEP-disturbance flux) of CMIP6 in the historical period to compare with GCP values and discussed whether the inclusion of the disturbance flux would change the results of this study. We found that the essential results and conclusions was not changed, although the numerical values would be shifted. We included the NBP-related results in the main text and the supplementary figures. Essentially, we found CMIP6 models might underestimate global NBP and will likely continue to underestimate it in the future, considering the expected more disturbances in a warming climate.

Personally, I also believe a manuscript such as this that discusses future carbon budgets would benefit from a historical comparison to data. This may be beyond the scope, but how should readers interpret the large differences between CMIP6 models and the GCP, and what does this ultimately mean for the future projections? Do models that tend to represent historical properties better (e.g., upscaled carbon fluxes from FLUXCOM, global biomass, LAI, etc.) tend to project higher or lower carbon sinks? These are important questions, some of which could be addressed in the Discussion and brining in past literature. Along similar lines, the Discussion could benefit from more context and interpretation. Why haven't uncertainties been reduced throughout CMIP versions, how would including permafrost carbon and disturbances change the assessment, what are the major recommendations on ways forward considering this?

Response: Thanks for the insightful suggestions. We extended our discussions on the historical comparisons by using the GCP data. In fact, we tried to use FLUXCOM NEP product to benchmark the CMIP6 models, but we finally decided to not include the results in the paper because of the large uncertainty of FLUXCOM NEP in the high latitudes (personal discussion with the developer of FLUXCOM; pasted below). We agree that LAI is a useful variable for understanding the uncertain CMIP6 carbon fluxes. Thus, we added discussion on CMIP6-estimated LAI, which has been suggested to be biased with high uncertainty by multiple other independent studies. These additional discussions provide our understandings of the uncertainties throughout CMIP versions and the recommendations for the future modeling efforts.

"Dear Han,

no, no conversion factor needed. It's unfortunately true that the mean NEE is not realistic. It seems to be mainly a problem of biased flux tower data in the tropics. It is discussed here.

cheers,

martin

"

[Figure]

Figure. Unrealistically high NEP from FLUXCOM.

Why did the authors analyze land surface temperature as opposed to a property such as 2 m air temperature, which is much more commonly used as a metric and benchmark? Land surface temperature accounts for not only the climate changes but also land surface responses, and in that way seems to add unnecessary complexity.

Response: Thanks for the suggestion. We used 2-m air temperature as recommended in the revised manuscript, which brings minimal changes to the results and conclusions. All analyses and figures accordingly were updated in the revised manuscript.  Please see updated Fig. S2, Table S2 and Fig. 2.

Finally, the manuscript would benefit from a thorough grammar check throughout. I addressed a minor fraction of grammatical errors in my comments below, but many more remain.

Response: We have gone through the manuscript again and corrected the grammar mistakes.

Technical corrections

L 16: What's the meaning of the word 'extent' here - spatial distribution, magnitude, or other?

Response: We meant magnitude. It is corrected in the revised manuscript.

L 20: do you mean to say CMIP5 here?

Response: No, here C4MIP is "Coupled Climate-Carbon Cycle Model Intercomparison Project" (Friedlingstein et al., 2006) and CMIP5 is "Coupled Model Intercomparison Project Phase 5" (https://pcmdi.llnl.gov/mips/cmip5/index.html).

Friedlingstein, P., Cox, P., Betts, R., Bopp, L., von Bloh, W., Brovkin, V., Cadule, P., Doney, S., Eby, M., Fung, I., Bala, G., John, J., Jones, C., Joos, F., Kato, T., Kawamiya, M., Knorr, W., Lindsay, K., Matthews, H. D., Raddatz, T., Rayner, P., Reick, C., Roeckner, E., Schnitzler, K.-G., Schnur, R., Strassmann, K., Weaver, A. J., Yoshikawa, C., & Zeng, N. (2006). Climate–Carbon Cycle Feedback Analysis: Results from the C4MIP Model Intercomparison, *Journal of Climate*, *19*(14), 3337-3353.
https://journals.ametsoc.org/view/journals/clim/19/14/jcli3800.1.xml

L20: NHL was defined and used previously as a plural noun, but here singular. Please remain consistent.

Response: we corrected this mistake in the revised manuscript.

-Maybe define what domain NHL is referring to in the abstract?

Response: we added the definition of NHL (poleward of 50$^{o}$N) in the revised manuscript.

L 32: The land carbon sink changes by a large amount interannually owing to annual climate oscillations, disturbances, etc.

Response: We revised the sentence as "releases a similar amount of carbon back to the atmosphere through respirations from plant metabolism and microbial activities (i.e., autotrophic and heterotrophic respirations) in response to climate oscillations and disturbances-induced emissions, resulting in a land carbon sink of about 3.4 Pg C/year."

L 45: this reference is almost a decade old now, and the literature it cites is over a decade; consider adding newer references for warming rates

Response: We updated the sentence with an up-to-date reference.

During the last few decades, the temperature in northern high-latitudes (NHL, poleward of 50 °N) regions has been rising fast. The Arctic Circle (66.5-90 °N) has warmed more than 0.7 °C per decade since 1979, almost four times faster than the globe (Rantanen et al., 2022).

Rantanen, M., Karpechko, A.Y., Lipponen, A. *et al.* The Arctic has warmed nearly four times faster than the globe since 1979. *Commun Earth Environ* **3**, 168 (2022). https://doi.org/10.1038/s43247-022-00498-3

L 76: What does the word 'devoted' mean here?

Response: we meant the ScenarioMIP was designed to understand climate change in the future scenarios. We removed the 'the most devoted MIP' to avoid confusion.

L 77: missing parenthesis

Response: Corrected.

L 86: Can the authors expand on the 'newly updated data' they're referring to?

Response: The 'newly updated data' are the datasets such as population and GDP that are used to drive Integrated Assessment Models (IAMs) to generate the SSP scenarios. But it might not be accurate here. We changed it to 'new conceptual design of future societal development and evolution with different assumptions on the challenges to mitigation and adaptation to the climate change' in the revised manuscript.

L 114: Incorrect use of "i.e.". Could say "…in this study, including…"

Response: Corrected.

L 126: Did the authors account for non-land fractions of grid cells in their area-weighting?

Response: non-land fractions of grid cells were not accounted in the area-weighting. We clarified this point in the revised manuscript.

L 138. The readers would benefit from more details on how these sensitivity analyses were conducted

Response: The sensitivity analyses were performed by calculating the relative changes in carbon fluxes to their current levels (represented by the mean of 2010-2015) in response to the temperature rises at an increment of 1°C (Pg C/°C) or atmosphere $CO_2$ concentration at an increment of 1 ppm (Pg C/ppm) for each model at both the global and NHL scales.

Figure 1: 2095-2100 is a short period to use to calculate standard deviations. More typical would be something like a 20 year time period

Response: We updated the error bars in Figure 1 by using the numbers over 2081-2100.

L 189-190: Change 'huge' to something like 'large'

Response: all 'huge' has been changed to 'large'.

Fig 1: Difficult to see the GCP values with all the light blue bars. Possible to change that into a shaded time series as well?

Response: we used shaded time series and changed the color of GCP values. Please see updated Fig. 1.

L 273: 'Minimization'? Do the authors mean mineralization?

Response: Corrected.

L 281: What does the word 'special' mean here?

Response: we meant the NHL was different from the other regions in terms of carbon cycling processes. We changed the word to "complicated".

L 287: The carbon balance will also significantly be impacted by disturbances, mentioned in the introduction but mostly not included in the CMIP6 models. This point should be emphasized.

Response: Thanks for pointing this issue out. We revised the sentence to "Besides the disturbance-induced carbon loss, the carbon balance in the terrestrial ecosystems will be

determined by the difference between rising primary productivity and the accelerated soil carbon decomposition driven by the interplay of multiple climate drivers (McKane et al., 1997; Sistla et al., 2013)" to emphasize that disturbances-induced carbon loss is not included.

L 291: Particularly poor grammar

Response: we corrected the sentence as 'The global and NHL NEP are growingly positive in the future, but at lower rates than NPP and RH.' in the revised manuscript.

L 302: "plant functional types"

Response: Corrected.

L 305: What is meant by 'compensation' here?

Response: the 'compensation' here means the compensation for the effects of nitrogen limitation. We added more details in this sentence to avoid confusion.

L 332: Hyphen after Northern not necessary

Response: Corrected.

---

## Author Response (AR2)

I think the authors did a nice job addressing the reviewer comments. I only have one remaining minor question relating to the added discussion of CMIP6 models and LAI observations; that is, in the Discussion it was not clear to me if CMIP6 models are generally overestimating or underestimating observed LAI. Otherwise the manuscript could still probably benefit from an editorial grammar check. But I would suggest to publish after these minor revisions.

Response: we add the following sentence to complete the discussion as suggested by the reviewer: 'Moreover, they revealed that most of the CMIP6 models overestimated the LAI in non-forested vegetation areas against observations, which largely contributed to the general overestimation of the global mean LAI.' Please see line 311-312